# Early Motor Repertoire of Very Preterm Infants and Relationships with 2-Year Neurodevelopment

**DOI:** 10.3390/jcm11071833

**Published:** 2022-03-25

**Authors:** Amanda K.-L. Kwong, Roslyn N. Boyd, Mark D. Chatfield, Robert S. Ware, Paul B. Colditz, Joanne M. George

**Affiliations:** 1Queensland Cerebral Palsy and Rehabilitation Research Centre, Child Health Research Centre, Faculty of Medicine, The University of Queensland, Brisbane, QLD 4101, Australia; amanda.kwong@mcri.edu.au (A.K.-L.K.); r.boyd@uq.edu.au (R.N.B.); m.chatfield@uq.edu.au (M.D.C.); 2Clinical Sciences, Murdoch Children’s Research Institute, Melbourne, VIC 3052, Australia; 3Department of Physiotherapy, University of Melbourne, Parkville, VIC 3010, Australia; 4Australian Cerebral Palsy Clinical Trials Network CRE, The University of Queensland, Brisbane, QLD 4101, Australia; 5Menzies Health Institutes Queensland, Griffith University, Brisbane, QLD 4222, Australia; r.ware@griffith.edu.au; 6University of Queensland Centre for Clinical Research, The University of Queensland, Brisbane, QLD 4072, Australia; p.colditz@uq.edu.au; 7Perinatal Research Centre, Royal Brisbane and Women’s Hospital, Brisbane, QLD 4029, Australia; 8Physiotherapy Department, Queensland Children’s Hospital, Children’s Health Queensland Hospital and Health Service, Brisbane, QLD 4101, Australia

**Keywords:** preterm infant, neurodevelopment, motor optimality score, general movements, motor impairment, cognitive impairment

## Abstract

The Motor Optimality Score, revised (MOS-R) is an extension of the Prechtl General Movements Assessment. This study aims to determine the relationship between MOS-R and 2-year neurodevelopmental outcomes in a cohort of 169 infants born very preterm (<31 weeks’ gestational age), and to examine the predictive validity of the MOS-R at 3–4 months’ corrected age (CA) above perinatal variables associated with poor outcomes, including Prechtl fidgety movements. Development at 2 years’ CA was assessed using Bayley Scales of Infant and Toddler Development, Third edition (Bayley-III) (motor/cognitive impairment: Bayley-III ≤ 85) and Neurological, Sensory, Motor, Developmental Assessment (NSMDA) (neurosensory motor impairment: NSMDA ≥ 12). Cerebral palsy (CP) was classified at 2 years as definite or clinical. The MOS-R was related to 2-year outcomes: Bayley-III motor (*B*_MOS-R_ = 1.24 95% confidence interval (0.78, 1.70)), cognitive (*B*_MOS-R_ = 0.91 (0.48, 1.35)), NSMDA scores (*B*_MOS-R_ = −0.34 (−0.42, −0.25)), definite CP (odds ratio [OR] 0.67 (0.53, 0.86)), clinical CP (OR 0.74 (0.66, 0.83)) for each 1-point increase in MOS-R. MOS-R ≤ 23 predicted motor (sensitivity 78% (60–91%); specificity 63% (54–72%)) and neurosensory motor impairment (sensitivity 86% (64–97%); specificity 59% (51–68%)). The MOS-R is strongly related to CP and motor and cognitive delay at 2 years and is a good predictor of motor and neurosensory motor impairment.

## 1. Introduction

Preterm birth affects one in 11 births in Australia. Of those, approximately 20% are born very preterm (VPT, <32 weeks’ gestation) and are more likely to experience developmental delay than infants who were born at term [1,2,3]. Potential delays in infants born VPT may be amenable to early intervention [4]. Early detection of developmental delay is a process of ongoing assessment and consideration of key perinatal events known to affect long-term outcomes [5,6]. In contrast, the prediction of cerebral palsy (CP) has been afforded much attention, with recent guidelines recommending the combination of brain neuroimaging, the Prechtl General Movements Assessment (GMA) [7], and neurological examination (Hammersmith Infant Neurological Examination) [8] for a diagnosis of high-risk CP at less than 6 months’ corrected age (CA) [9]. In order to direct early intervention resources to those most likely to experience developmental delay, it is paramount to accurately predict which infants born very preterm will go on to experience motor or cognitive delay.

The GMA is a widely used tool within Australian Neonatal Intensive Care Units (NICU) [10] but has lower predictive validity for cognitive and motor delays compared with CP [11,12,13,14]. This is particularly relevant for fidgety movements assessed according to Prechtl’s method [7] which are a pattern of movements characterized by continuous, small amplitude movements of moderate velocity which can be typically recognized throughout the limbs and body of an alert infant between 9 and 20 weeks’ corrected age. Fidgety movements are classified as normal, sporadic (observed only occasionally), absent (not seen at all), or abnormal (exaggerated with larger amplitude). A semi-quantitative measure that is an extension of the GMA, known as the Motor Optimality Score, revised 2019 (MOS-R) [15,16], provides an optimality score of an infant’s motor movements and patterns between 9 and 20 weeks’ CA. The MOS-R may provide a better association with later non-CP developmental outcomes [11,17,18,19,20,21,22] or indicate the severity of CP based on Gross Motor Function Classification System (GMFCS) grading [15], but while emerging evidence is encouraging, few studies have investigated the cut-off score to maximize predictive validity for motor, cognitive, or neurosensory impairment at 2 years’ CA. The MOS-R is conducted without assessor handling, a key advantage where face-to-face assessments may be limited, but requires additional advanced training beyond a basic general movements course. The clinical utility of the MOS-R, therefore, needs to be explored in the very preterm population to justify the resources needed to acquire additional training.

Infants born preterm experience higher rates of cognitive, language, and motor delay, which can later affect academic performance [2]. As such, early detection of developmental delays not associated with disability should also be investigated. The classification of impairment at 2 years by commonly used tests may fail to classify all children who experience developmental delay at 8 years of age [23]. Other tests that evaluate more subtle differences in development, such as the Neurological, Sensory, Motor, Developmental Assessment (NSMDA) [24] may provide a better measure of impairment in children who otherwise appear to demonstrate typical development on standardized testing.

The current study aims to determine the relationships between the MOS-R and cognitive, motor, neurosensory motor, or CP outcomes at 2 years’ CA. Additionally, the study aims to investigate whether the MOS-R is independently related to 2-year outcomes after adjustment for Prechtl fidgety movements or variables known to affect long-term outcome among infants born preterm <31 weeks’ gestation. Third, the study aims to determine if the MOS-R, which is inclusive of fidgety movements assessment, provides better predictive ability with 2-year motor, cognitive, neurosensory motor, or CP outcomes than fidgety movements alone. Further to this, the predictive values of motor, cognitive, and neurosensory motor impairment will be described for the best cut-off point of the MOS-R.

## 2. Materials and Methods

### 2.1. Participants

Participants included infants born very preterm, between February 2016 and December 2018, who were free from congenital anomalies from the Prediction of Preterm Brain Outcomes (PREBO) [25]. Infants were recruited prospectively from the Royal Brisbane Women’s Hospital (RBWH). Infants recruited to an earlier study cohort (Prediction of Preterm Motor Outcome (PPREMO)) [3] born January 2013–February 2016 at the RBWH were also invited to participate in the PREBO study. Families of the infants lived within 200 km of their recruiting hospital and needed to be able to communicate in English. Included infants were free from congenital or chromosomal abnormalities known to affect neurodevelopment. Infants with any contraindication to magnetic resonance imaging (MRI) were excluded due to the MRI component of the overarching study. Parents of infants provided written informed consent to participate, and the study was approved by the Human Research Ethics Committees at the Royal Brisbane Women’s Hospital (HREC/15/QRCH) and The University of Queensland (2015000290) and registered on the Australian New Zealand Clinical Trial Registry (ACTRN12615000591550). The studied cohort [3,25] presents those with completed 2-year data.

### 2.2. Acquisition and Scoring of General Movements and Motor Optimality Score, Revised

The MOS-R was applied to video recordings of infants by an advanced-certified assessor (AK), masked to infant clinical history. The GMA component of the MOS-R was scored by two advanced-certified raters and any disagreements were settled through review between three raters. Reliability for the MOS-R has previously been reported to be high with intraclass correlation coefficients of 0.86–0.98 [26,27,28,29]. The MOS-R consists of five subsections: (i) fidgety movements, which is assigned a score of 1, 4 or 12 for absent/sporadic, abnormal or normal movements respectively; (ii) observed movement patterns, that tallies the number of typical and atypical observed postures and assigns a score of 1 for more atypical movements, 2 for equal number of atypical and typical movements, and 4 for more typical movements/postures; (iii) the age-adequate movement repertoire scores 1 (absent), 2 (reduced), or 4 (present) which is determined by a checklist of movements that are expected to be observed depending on the age of assessments: at 9–11, 12–13, 14–15, and over 16 weeks’ CA; (iv) observed postural patterns, which follows a similar scoring concept to subsection (ii), but observes postural patterns rather than movement patterns; (v) movement character, which describes an infant’s movements as either smooth and fluent (score 4), not smooth and fluent, but not cramped-synchronized (score 2), or cramped-synchronized (score 1). Each subsection was scored and added to produce a final continuous score between 5 and 28 [15].

For this article (with the exception of Appendix A), GMA was classified as absent/abnormal (MOS-R, first subsection score 1 or 4, with sporadic movements classified as “absent”) or as normal fidgety (MOS-R, first subsection score 12).

### 2.3. Assessment of 2-Year Outcomes

Motor and cognitive outcomes were assessed by a masked pediatric physiotherapist using the Bayley Scales of Infant and Toddler Development, 3rd edition (Bayley-III) [30] and the NSMDA [24] described in a previously published protocol [3]. As the Bayley-III is known to underestimate the extent of delay in Australian children [31], a cut-off of Bayley composite score ≤85 to indicate impairment for the motor and cognitive domains was used. The NSMDA was used as a continuous functional grade score (higher scores indicate worse impairment) and neuromotor-sensory impairment was defined as an NSMDA functional score of 12 or above.

CP was assessed at 2 years’ CA by pediatricians who administered standardized neurological assessments and classified CP using the GMFCS [32] as well as CP motor distribution [33], and as described in previous protocols/study [3,25]. CP was classified as definite, possible (where children had neurological abnormalities, but could not be clearly classified as CP), or no CP. Infants were dichotomized as having definite CP (versus possible or no CP) or having a clinical CP diagnosis (definite or possible).

Social risk was determined through parent questionnaire which surveyed family structure, education level of primary caregiver, occupation of primary income earner, employment status of primary income earner, language spoken at home, and maternal age at birth with each category receiving 0–2 points depending on social risk, then tallied in accordance with Roberts et al. (2008) [34]. Families were considered to have a higher social risk if their score was 2 points or more and lower social risk if their score was 0–1 [34].

### 2.4. Statistical Analysis

Associations with Bayley-III motor and cognitive continuous standard scores and NSMDA functional scores were assessed using linear regression. Associations with CP (definite and clinical classifications) were assessed using logistic regression. Infants with any missing data were not included in the analysis.

First, univariable associations between MOS-R and GMA (absent/abnormal fidgety movements) and 2-year outcomes were reported. The average change in the outcome per one-unit increase for the MOS-R (*B*_MOS-R_) and for absent/abnormal and normal fidgety movements (*B*_GMA_) was reported. Subsequent multivariable analyses using two different models were conducted (i) multivariable model 1: Predictors were MOS-R and absent/abnormal fidgety movements; (ii) multivariable model 2: Predictors were MOS-R and the following variables known to influence long-term outcomes: [5,6] brain injury (being the presence of major [grade 3 or 4] intraventricular hemorrhage or cystic periventricular leukomalacia), the administration of postnatal corticosteroids, male sex and higher social risk, or lower gestational age. For the analysis of CP, gestational age was the only variable within the model due to the low rate of CP.

Receiver operating characteristic (ROC) curves were plotted for MOS-R and absent/abnormal fidgety movements for each of the motor, cognitive, and neurosensory motor impairment and CP outcomes. Liu [35] and Youden’s [36] statistics were used to determine the optimal cut-off point for the best combination of sensitivity and specificity values. The predictive validity of the MOS-R to detect motor, cognitive, and neurosensory motor impairment and CP was described by calculating the Area Under ROC Curve (AUC), sensitivity, specificity, positive and negative predictive values, and accuracy (percentage of correctly classified cases). Data were analyzed using Stata (StataCorp. 2019. Stata Statistical Software: Release 16. College Station, TX, USA: StataCorp LLC.).

## 3. Results

### 3.1. Participants and Characteristics

Two-hundred and eighty-eight infants were recruited to the PREBO and PPREMO studies and 216 infants were assessed using the MOS-R. To date, 169 infants have been followed up at 2 years’ CA for the Bayley-III motor domain (160 participants, 56%), Bayley-III cognitive domain (169 participants, 59%), and NSMDA (162 participants, 56%), and had a CP assessment (159 participants, 55%) (Figure 1). The characteristics of infants assessed are described in Table 1. There was a lower proportion of families with high social risk and a higher rate of postnatal corticosteroid use among those with complete data (Appendix A). There were no other strong differences between groups who had complete and incomplete data. The distribution of the MOS-R across infants with absent, abnormal, and normal fidgety general movements is reported in Appendix A and the proportion of scores across the MOS-R’s five subsections is reported in Figure 2.

### 3.2. Relationships between MOS-R and 2-Year Outcomes

The univariable and multivariable relationships between MOS-R and 2-year outcomes are reported in Table 2. The MOS-R was strongly related to Bayley-III motor, cognitive, and NSMDA scores and CP outcome (both definite and possible/clinical diagnoses), and the MOS-R explained the highest proportion of variance for the NSMDA compared with motor or cognitive Bayley-III scores. The relationships remained strong for Bayley-III motor scores and NSMDA functional scores when corrected for absent/abnormal fidgety movements (model 1). The odds ratio for the likelihood of CP from GMA could not be calculated as all infants with CP had absent/abnormal fidgety movements and no infants with CP had normal fidgety movements. The MOS-R remained strongly related to motor, cognitive Bayley-III scores, NSMDA scores, and CP outcome (both definite and clinical CP) even when adjusted for perinatal variables (model 2). The univariable and multivariable relationships between absent/abnormal fidgety movements and 2-year outcomes are reported in Table 3.

### 3.3. Predictive Validity of MOS-R and 2-Year Outcomes

The MOS-R had excellent prediction of CP outcome, with moderate–good results for motor and neurosensory motor impairment (Figure 3A–E). The ideal MOS-R cut-off for the prediction of motor, cognitive, and neurosensory motor impairment was scores ≤23 and for CP, ≤15 with predictive values listed in Table 4. The AUC was higher for MOS-R than GMA for the prediction of motor (difference in AUC, *p* = 0.002) and neurosensory motor impairment (difference in AUC, *p* = 0.005) (Figure 3A–E), however, the AUC values for motor and neurosensory motor impairment were moderate–good. Overall, sensitivity and positive predictive values of the MOS-R for motor, cognitive, and neurosensory motor impairment were higher than the sensitivity and positive predictive values of the GMA fidgety movement score to predict the same outcomes, but specificity and accuracy were lower for the MOS-R than the GMA in predicting motor, cognitive, and neurosensory motor impairment. Both MOS-R and GMA had good–excellent negative predictive values. Predictive values of the MOS-R and GMA did not differ for CP outcome.

## 4. Discussion

A strong relationship was observed between the MOS-R and neurodevelopment as determined by motor and cognitive Bayley-III, NSMDA scores, and CP diagnosis in this prospective cohort of Australian infants born VPT. Furthermore, the MOS-R contributes independently to motor and neurosensory motor outcomes in addition to the assessment of fidgety movements alone, and independently from other perinatal variables known to be associated with poorer outcomes. While the MOS-R was strongly related to cognitive outcome on univariable analysis, the relationship weakened when corrected for absent/abnormal fidgety movements, indicating that the MOS-R was not independently related to cognitive outcomes when corrected for absent/abnormal fidgety movements. The relationship between the MOS-R and the NSMDA explained a higher proportion of variance than Bayley-III motor or cognitive scores. The MOS-R also had better predictive validity for non-CP developmental outcomes than absent/abnormal fidgety movements but the MOS-R was no better than absent/abnormal fidgety movements for predicting CP.

Infants born VPT who experience developmental delay are overwhelmingly represented by those with a spectrum of mild–severe delays in motor, cognitive, or language function, and less so with CP [2,37]. Furthermore, the rate of motor impairment appears to be increasing in other Australian cohorts of extremely preterm infants over time [38]. Therefore, the early detection of non-CP developmental delay is warranted and accordingly, this study’s findings indicate that the MOS-R can provide additional insight into early neurodevelopmental delay. The MOS-R, having a better relationship with impairment outcomes than absent/abnormal fidgety movements, may be an early functional manifestation of injury or represent an early sign of the interplay between cognitive and motor development [39], and may be used as an early indicator of developmental delay.

Similar studies have also found comparable results to the current study. A study based on infants born VPT in the US showed that the MOS-R was independently related to Bayley-III motor, cognitive, and language outcomes in a multivariable model with the GMA [18]. Within the current study, the relationship of the MOS-R represented a much higher proportion of variance with the NSMDA than Bayley-III scores, suggesting that the MOS-R is better related to the NSMDA at 2 years’ CA. Consistent associations between the MOS-R and 2-year outcome have also been noted in extremely preterm (<28 weeks’ gestational age) populations where lower MOS-R assessed at 14–16 weeks’ corrected age has been related to motor and cognitive outcomes [29]. The MOS-R has also been found to have a strong relationship with other motor assessments, with one study finding a strong relationship between the MOS-R and 1-year Peabody Developmental Motor Scales outcomes [19]. Additionally, another study which investigated predictive values for adverse outcomes (neurosensory disability, CP, or Movement Assessment Battery for Children score ≤5th percentile) noted a cut-off MOS-R of 21 among infants born extremely preterm [22].

Within the current study, absent/abnormal fidgety movements alone also had strong relationships with motor, cognitive, and neurosensory motor outcomes, but the MOS-R had better prediction of impairment outcomes based on the set Bayley-III or NSMDA thresholds. MOS-R scores ≤23 had a lower rate of false negatives than the GMA, reflected in the higher sensitivity of the MOS-R to detect developmental delay, but a much higher rate of false positives. In most instances, infants who do have fidgety movements but poorer MOS-R will eventually have a typical outcome, and given the high rate of false positives, using the MOS-R to detect non-CP developmental delay may result in over-surveillance of these infants with normal fidgety movements and has potential to cause unnecessary anxiety in parents and/or overuse of resources. Conversely, clearing infants with normal fidgety movements but poorer MOS-Rs from further surveillance may miss some infants who will later go on to have another non-CP neurodevelopmental delay which can have a significant impact on participation in daily life, and potentially lose opportunities to engage in timely early intervention. One solution to overcoming inaccurate early diagnoses of developmental delay would be to consider the MOS-R alongside other standardized neurological assessments. Indeed, future studies may need to contrast the relationships of the MOS-R with other assessments, for example, the Test of Infant Motor Performance [40] or the Hammersmith Infant Neurological Examination [8,41].

Other studies, including multiple systematic reviews [42,43], have agreed that the assessment of fidgety movements according to the Prechtl GMA has high predictive validity for CP, but a study in a clinical population has found lower predictive validity of the Prechtl GMA to detect CP [44]. Our current study defined CP in two ways: the first, which was aligned with previous research methods, where only infants with definite CP were included for prediction, and the second, a clinical CP diagnosis, where some infants may not show clear signs of CP, but had clinical markers that leaned towards a possibility of CP. Our findings reflect trends from both clinical and research settings, with the classification of CP from research settings producing a higher predictive value. CP will need to be confirmed with GMFCS classification at a later age (i.e., 5 years of age or older) according to Australian CP register guidelines [45], and further assessment at 6 years is planned for the current cohort [25]. This study supports the notion that the GMA has better prediction for CP than neurodevelopmental delay [12,13,14] and that the MOS-R may provide the extra information needed in addition to fidgety movements alone to detect motor, cognitive, or neurosensory motor impairment.

### 4.1. Strengths

A strength of the study is the relatively large sample size of infants born very preterm, recruited prospectively, representing one of the larger cohorts of infants studied using the MOS-R. Assessors were also masked to clinical history at all stages of assessment, and adjustment for key perinatal variables also strengthened the study. Additionally, our study used an a motor assessment (NSDMA) in addition to the Bayley-III to provide an alternative and in-depth measure of neurosensory motor performance due to the Bayley-III having only moderate relationships with later motor outcomes [46].

### 4.2. Limitations

We were unable to compare the findings or measure impairment with a term-reference group. A conscious decision to recruit infants born VPT only was made due to resource limitations and available existing evidence [31,47] to determine appropriate cut-off scores for level of impairment within an Australian population. This study was also unable to ascertain the level of engagement with early intervention, which may have affected the rate of false positives of the MOS-R to predict motor, cognitive, and neurosensory motor impairment. Assessment at two years is also early within the trajectory of a child’s development. As such, participants are currently being assessed at six years of age and repeated analysis may identify with better accuracy the predictive validity of the MOS-R for older childhood outcomes [48]. Finally, due to resource constraints, only one assessor was available to score the MOS-R, however, previous studies have indicated high inter-rater reliability [29].

### 4.3. Future Considerations

While the relationship between the MOS-R and neurodevelopmental outcomes is strong within the current population, further studies are needed in other high-risk populations and within other geographical settings to ensure that the findings can be applied to other infant risk groups. Additionally, larger studies focused on recruiting infants at high risk of developing CP are needed to ascertain prospectively the relationship between MOS-R and GMFCS level for infants who go on to develop CP. Finally, the relationships between MRI assessments and MOS-R should also be explored to guide early screening for developmental delay and to understand the structural/neurological correlates of MOS-R compared with the GMA.

## 5. Conclusions

The MOS-R is strongly related to neurodevelopmental outcomes, independent of fidgety movements alone and other perinatal variables associated with poor outcomes. The MOS-R has better prediction of developmental impairment than the assessment of fidgety movements, moderate prediction of motor and cognitive outcomes, and good prediction of neurosensory motor outcomes. Care should be taken to ensure that the MOS-R is still used alongside other assessments to guide clinical care.

## Figures and Tables

**Figure 1 jcm-11-01833-f001:**
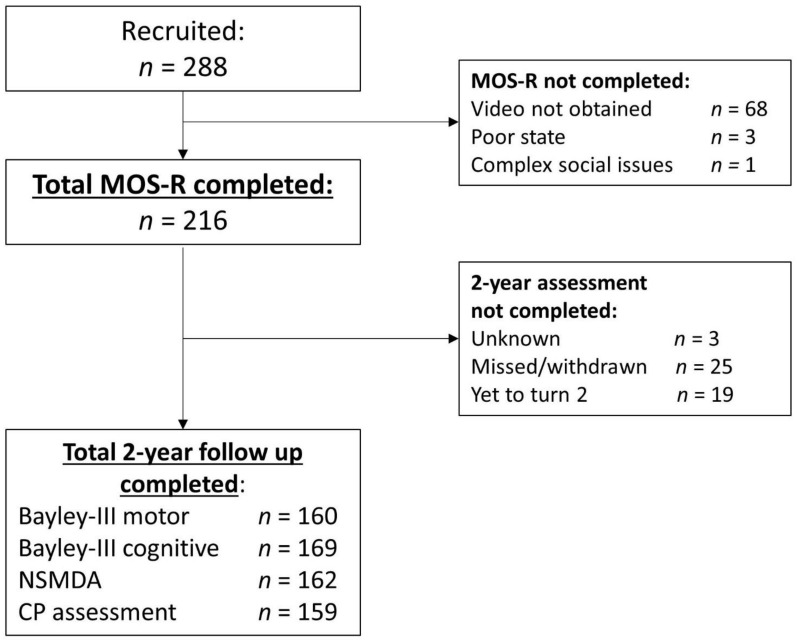
Flowchart of participants. Abbreviations: MOS-R: Motor Optimality Score, revised; Bayley-III: Bayley Scales of Infant and Toddler Development, 3rd edition; NSMDA: Neurological, Sensory, Motor Developmental Assessment; CP: cerebral palsy.

**Figure 2 jcm-11-01833-f002:**
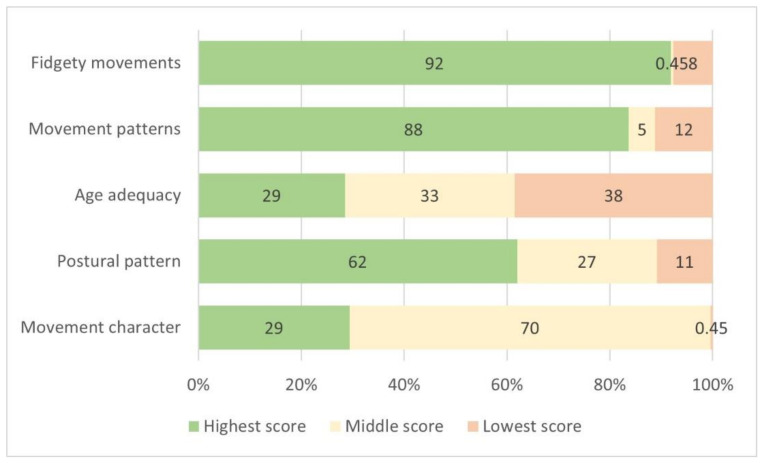
Proportion of infants represented within each subsection of MOS-R.

**Figure 3 jcm-11-01833-f003:**
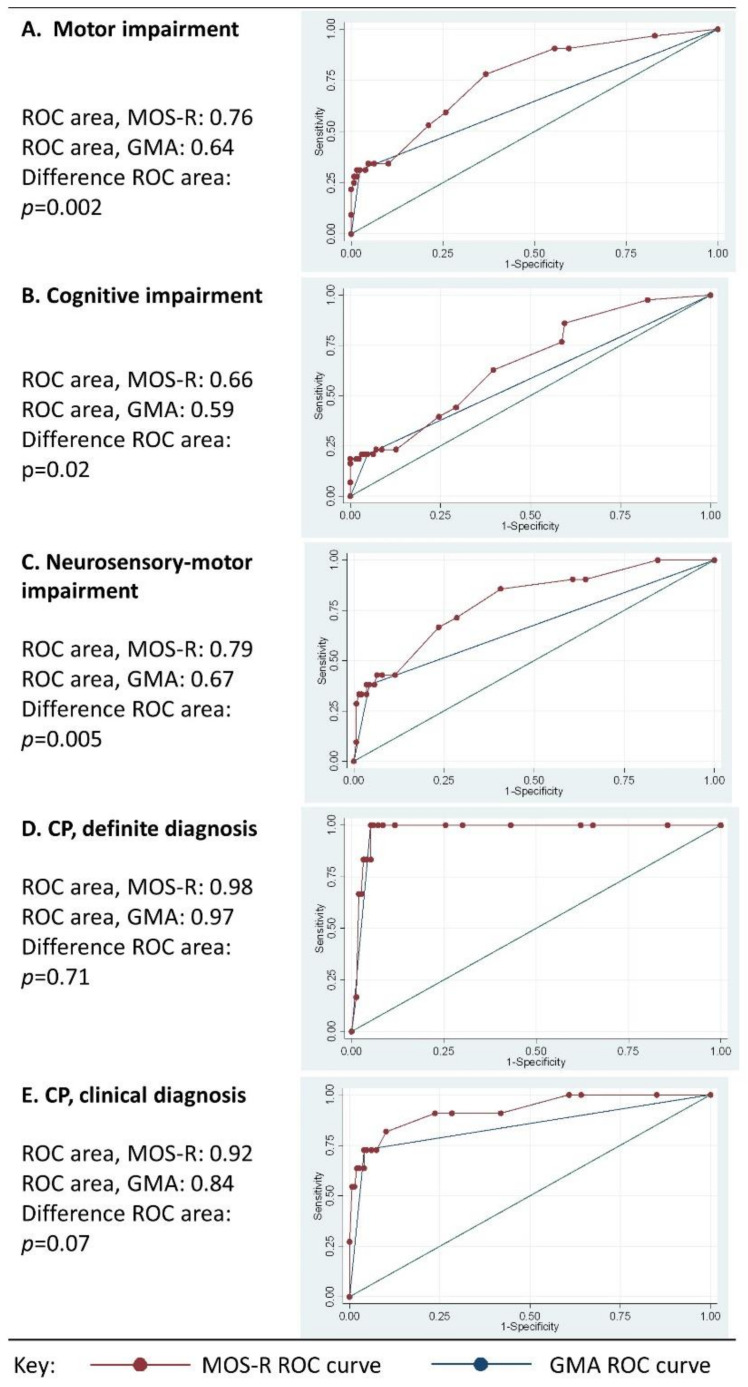
(**A**–**E**): ROC curves for several outcomes with GMA and MOS-R, testing hypothesis that AUC is not the same. MOS-R cut-off ≤ 23 for Figure 3A–C, MOS-R cut-off ≤ 15 for Figure 3D,E; Abbreviations: ROC: receiver operating characteristic; MOS-R: motor optimality score, revised; AUC: area under curve.

**Table 1 jcm-11-01833-t001:** Perinatal characteristics of infants assessed at 3–4 months and 2 years.

Perinatal Details	MOS-R and 2-Year Follow-Up Completed
	*n* = 169
Gestational age at birth weeks, mean (SD)	28.6 (1.9)
Birth weight g, mean (SD)	1097 (310)
Males	95 (56%)
Multiple births	54 (32%)
Higher social risk	81/168 (48%)
IVH grade 3 or 4	16 (9%)
Periventricular leukomalacia	8 (5%)
Postnatal corticosteroids	30 (18%)
Bronchopulmonary dysplasia	65 (38%)
3–4 month assessment	
Corrected age of GMA/MOS-R assessment, weeks, mean (SD)	13.4 (1.3)
MOS-R, median (IQR)	24.0 (21.0–26.0)
GMA	
Normal	154 (91%)
Abnormal	1 (0.6%)
Absent	14 (8%)
2-year details	
Age of assessment months, mean (SD)	24.2 (0.8)
CP	
No CP	148/159 (94%)
Possible CP	5/159 (3%)
Definite CP	6/159 (4%)
Bayley-III motor composite score, mean (SD) *n*	97.9 (16) *160*
Motor impairment	32/159 (20%)
Bayley-III cognitive composite score, mean (SD) *n*	96.5 (15.1) *169*
Cognitive impairment	43 (25%)
NSMDA functional grade, mean (SD) *n*	8.5 (2.9) *161*
Neurosensory motor impairment	21/161 (13%)

Data are *n* (%) unless otherwise specified. Abbreviations: MOS-R: motor optimality score, revised; SD: standard deviation; IVH: intraventricular hemorrhage; GMA: General Movements Assessment; CP: cerebral palsy; NSMDA: Neurological, Sensory, Motor, Developmental Assessment.

**Table 2 jcm-11-01833-t002:** Association between MOS-R with Bayley-III motor, cognitive scores, NSMDA functional grades and CP (definite and clinical classifications).

Motor Optimality Score, Revised				
	Model	*B*_MOS-R_ (95% CI)	*p*-Value	R^2^
Bayley-III motor scores	univariable	1.24 (0.78, 1.70)	<0.001	0.15
	multivariable 1	0.97 (0.12, 1.83)	0.03	0.15
	multivariable 2	1.22 (0.76, 1.69)	<0.001	0.28
Bayley-III cognitive scores	univariable	0.91 (0.48, 1.35)	<0.001	0.09
	multivariable 1	0.63 (−0.21, 1.47)	0.13	0.10
	multivariable 2	−0.34 (−0.42, −0.25)	0.001	0.23
NSMDA functional scores *	univariable	−0.34 (−0.42, −0.25)	<0.001	0.28
	multivariable 1	−0.28 (−0.43, −0.12)	0.001	0.28
	multivariable 2	−0.30 (−0.39, −0.21)	0.001	0.37
		**Odds Ratio (95% CI)**		
Definite CP	univariable	0.67 (0.53, 0.86)	0.002	n/a
	multivariable 1	**	**	
	multivariable 2 ^†^	0.68 (0.53, 0.86)	0.001	
Clinical CP	univariable-MOS-R	0.74 (0.66, 0.83)	<0.001	
	multivariable 1	0.68 (0.48, 0.96)	0.03	
	multivariable 2 ^†^	0.74 (0.66, 0.83)	0.001	

Abbreviations: MOS-R: Motor Optimality Score, revised 2019; Bayley-III: Bayley Scales of Infant and Toddler Development, 3rd edition; NSMDA: Neurological, Sensory, Motor Developmental Assessment; CP: cerebral palsy; Multivariable 1: MOS-R adjusted for fidgety movements, for given 2-year outcome; Multivariable 2: MOS-R adjusted for sex, postnatal corticosteroids, high social risk, brain injury (IVH ¾ or cystic PVL), gestational age; *B*_MOS-R:_ average change in Bayley-III/NSMDA score for every 1-point increase in MOS-R score; *: NSMDA, higher scores indicate poorer function; ** unable to calculate odds ratio due to diverging estimates; ^†^ adjusted for gestational age only.

**Table 3 jcm-11-01833-t003:** Association between absent/abnormal fidgety movements with Bayley-III motor, cognitive scores, NSMDA functional grades, and CP (definite and clinical classifications).

General Movements Assessment–Absent/Abnormal Fidgety Movements				
	*B*_GMA_ (95% CI)		*p*-Value	R^2^
Bayley-III motor scores	−20.55 (−28.99, −12.10)		<0.001	0.13
Bayley-III cognitive scores	−15.43 (−23.22, −7.63)		<0.001	0.08
NSMDA functional scores *	5.34 (3.79, 6.89)		<0.001	0.23
	**Odds ratio (95% CI)**			
Definite CP	*n*(CP)/*n* (absent/abnormal fidgety) 6/14	*n*(CP)/*n* (normal fidgety) 0/145	<0.001 ^‡^	n/a
Clinical CP	63.1 (13.3, 299.8)		<0.001	

Abbreviations: Bayley-III: Bayley Scales of Infants and Toddler Development, 3rd edition; NSMDA: Neurological, Sensory, Motor, Developmental Assessment; CP: cerebral palsy; *B*_GMA_: average difference in Bayley-III composite score or NSMDA score for infants with absent/abnormal fidgety movements; *: NSMDA, higher scores indicate poorer function; ^‡^: Fisher’s exact test.

**Table 4 jcm-11-01833-t004:** Diagnostic statistics for MOS-R optimal cut-off and GMA with motor, cognitive, or neurosensory motor impairment.

	Sensitivity	Specificity	Positive Predictive Value	Negative Predictive Value	Accuracy (Correctly Classified)
MOS-R *					
Motor impairment	78% (60–91%)	63% (54–72%)	77% (46–95%)	85% (78–90%)	66% (58–74%)
Cognitive impairment	63% (47–77%)	60% (51–69%)	60% (32%, 84%)	78% (71–84%)	61% (53–68%)
Neurosensory motor impairment	86% (64–97%)	59% (51–68%)	57% (29–82%)	91% (85–95%)	63% (55–70%)
Cerebral palsy (definite only)	100% (54–100%)	95% (90–98%)	43% (18–71%)	100% (98–100%)	95% (90–98%)
Cerebral palsy (possible and definite)	73% (39–94%)	96% (91–99%)	57% (29–82%)	98% (94–100%)	94% (90–97%)
GMA **					
Motor impairment	31% (16–50%)	98% (93–100%)	35% (24–47%)	92% (84–97%)	84% (78–90%)
Cognitive impairment	21% (10–36%)	95% (90–98%)	35% (25–47%)	83% (73–90%)	76% (69–83%)
Neurosensory motor impairment	38% (18–62%)	96% (91–98%)	24% (15–35%)	97% (90–99%)	88% (82–93%)
Cerebral palsy (definite only)	100% (54–100%)	95% (90–98%)	43% (18–71%)	100% (98–100%)	95% (90–98%)
Cerebral palsy (possible and definite)	73% (39–94%)	96% (91–99%)	57% (29–82%)	98% (94–100%)	94% (90–97%)

All data are % (95% CI). Abbreviations: CI: confidence interval; MOS-R: motor optimality score, revised; GMA: General Movements Assessment; * Cut-off for MOS-R scores ≤ 23 indicate poorer MOS-R for motor, cognitive, and neurosensory impairment; ** cut-off ≤ 15 for cerebral palsy.

## Data Availability

The data that support the findings of this study are available on request from the corresponding author. The data are not publicly available due to privacy and ethical restrictions.

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
