# Peer review of "Early Motor Repertoire of Very Preterm Infants and Relationships with 2-Year Neurodevelopment"

_jcm, 2022, doi:10.3390/jcm11071833_

Round 1

Reviewer 1 Report

I would like to commend the authors for a thoroughly conducted study and a well written manuscript. I feel that the authors are at some points not acknowledging all original sources related to GMA; e.g. in the Introduction quoting the GMA-Manual. Also, the review by Einspieler et al (Front Psych 2016) on GMA and cognitive development is missing. The recent publication by Örtqvist et al (Early Hum Dev 2021) describes in details all items of the MOS-R in extremely preterm born infants and should be mentioned here as well.

Research questions are very well described though “Prechtl fidgety movements” might be an insider terminology. At least a reference should be given; I suggest to add the article by Einspieler, Peharz and Marschik 2016.

Method

Very well designed, conducted and described.

Results

Table 1: 2 years details / there is no alignment between left and right column (CP)    

Fig 2: is very helpful and nicely reported, thank you.

All results are well and thoroughly described.

Subchapter 3.2. Infants had normal fidgety movements instead of normal fidgety GMA 

Discussion

Lines 256/257 – please add that it is a confirmation of Einspieler et al JCM 2019; the MOS-R (especially in its subscales ii-v) had shown to predict the severity of CP (GMFCS); but a high risk for CP can be solely predicted from the absence of fidgety movements known since Prechtl et al, Lancet 1997.

Line 320: There are numerous studies showing the predictive validity of fidgety movements for CP; the two mentioned ones are just examples. This should be made clear.

Please add to the Limitations that a two-year-outcome is by far not the endpoint of development. Non-CP motor, cognitive and neurosensory impairments might be still transient at that age.

It was a great pleasure reading this article. Best of luck for the future endeavors.

Author Response

Response to Reviewer 1 Comments

Thank you to the reviewers for taking the time to review our manuscript. We supply our responses to the reviewer comments below. Please also note, some additional errors were rectified at this review, including changes to American English spellings of the words, e.g. “pediatric”, “standardized” and “dichotomized”. The symbols for the coefficients reported in the statistics section was also altered where BMOS-R should be italicised B and subscript MOS-R or GMA.

Point 1: I would like to commend the authors for a thoroughly conducted study and a well written manuscript. I feel that the authors are at some points not acknowledging all original sources related to GMA; e.g. in the Introduction quoting the GMA-Manual. Also, the review by Einspieler et al (Front Psych 2016) on GMA and cognitive development is missing. The recent publication by Örtqvist et al (Early Hum Dev 2021) describes in details all items of the MOS-R in extremely preterm born infants and should be mentioned here as well.

Response 1: We acknowledge that these references should be included and have included Einspieler et al. 2004 (GMA manual) at the first mention of the GMA (page 1, line 44) and have also included a reference for the HINE (page 1, line 45, Haataja et al. 1999).

We have also included the review by Einspieler et al (2016) to statements introducing the known link between GMA and cognitive outcome at page 2, lines 51 and 54.

The reference for Ortqvist et al. 2021 has been added at first mention of the MOS-R on page 2, line 52.

Point 2: Research questions are very well described though “Prechtl fidgety movements” might be an insider terminology. At least a reference should be given; I suggest to add the article by Einspieler, Peharz and Marschik 2016.

Response 2: Thank you for identifying this oversight. Fidgety movements have not been described in the manuscript and a description has been inserted at page 2, line 51. The reference for Einspieler 2016 is on page 2, line 51.

Point 3: Table 1: 2 years details / there is no alignment between left and right column (CP)

Response 3: Thank you for identifying the error in formatting. We have adjusted Table 1 so that column 1 is left aligned and subcategories of GMA and CP are clearly formatted. (page 6)

We have also taken the opportunity to adjust the formatting for Tables 2, 3 and 4.

Point 4: Subchapter 3.2. Infants had normal fidgety movements instead of normal fidgety GMA 

Response 4: Thank you for identifying this error. We have changed page 7, line 205 to “absent/abnormal fidgety movements and line 206 to “normal fidgety movements”.

Point 5: Lines 256/257 – please add that it is a confirmation of Einspieler et al JCM 2019; the MOS-R (especially in its subscales ii-v) had shown to predict the severity of CP (GMFCS); but a high risk for CP can be solely predicted from the absence of fidgety movements known since Prechtl et al, Lancet 1997.

Response 5: Due to the low prevalence of CP in this study, we were unable to determine if the MOS-R was related to the severity of CP. We have therefore mentioned this within the discussion and as a possible future direction for future study of the MOS-R:

Page 2, Line 59: “The MOS-R may provide a better association with later non-CP developmental outcomes,[13-18, X3] or indicate the severity of CP based on Gross Motor Function Classification System (GMFCS) grading.[12](Einspieler et al. 2019)

And

4.3, future considerations: page 12, line 341-343:

Additionally, larger studies focused on recruiting infants at high risk of developing CP are needed to ascertain prospectively the relationship between MOS-R and GMFCS level for infants who go on to develop CP.

Point 6: There are numerous studies showing the predictive validity of fidgety movements for CP; the two mentioned ones are just examples. This should be made clear.

Response 6: The references listed are both systematic reviews, and therefore reference many other studies within the systematic reviews. We have added clarity by indicating that these two studies are systematic reviews:

Page 11, line 306: “Other studies, including multiple systematic reviews, [39,40] have agreed that the assessment of fidgety movements according to the Prechtl GMA has high predictive validity for CP…”

Point 7: Please add to the Limitations that a two-year-outcome is by far not the endpoint of development. Non-CP motor, cognitive and neurosensory impairments might be still transient at that age.

Response 7: We agree and note that our current cohort are being assessed at 6 years’ corrected age. We have added to the limitations:

Page 12, line 336-339:

“Assessment at 2-years is also early within the trajectory of a child’s development. As such, participants are currently being assessed at 6 years of age and repeated analysis may identify with better accuracy the predictive validity of the MOS-R for older childhood outcomes.”

Reviewer 2 Report

The authors report of 169 infants’ general movements at three months of age and their 2-year outcome is a valuable contribution to the understanding of a possible relationship between early motor behavior and later motor and cognitive development. The study is well performed and clearly presented.

INTRODUCTION. The statement in line 62, 63 and 64 warrant a reference. If not, move to discussion.

MATERIAL and METHODS. Line 98 and 99 state that only one assessor scored the Motor Optimality Score-Revised. Does this also apply to GMA or were several assessors involved? A more detailed description of how the MOS-R assessment and the GMA were done will be needed. Even if the reliability of MOS-R is described to be high only one assessor must be regarded as a weakness of this study.  Line 115: Did any of the infants have exaggerated fidgety movements, or is that regarded as the same as abnormal fidgety?

RESULTS.  Line 196: What about the infants with sporadic fidgety movements?

Author Response

Response to Reviewer 2 Comments

Thank you to the reviewers for taking the time to review our manuscript. We supply our responses to the reviewer comments below. Please also note, some additional errors were rectified at this review, including changes to American English spellings of the words, e.g. “pediatric”, “standardized” and “dichotomized”. The symbols for the coefficients reported in the statistics section was also altered where BMOS-R should be italicised B and subscript MOS-R or GMA.

Point 1: INTRODUCTION. The statement in line 62, 63 and 64 warrant a reference. If not, move to discussion.

Response 1: we have clarified this statement and altered to:

Page 2, line 67-69: “Infants born preterm experience higher rates of cognitive, language and motor delay, which can later affect academic performance.[2] As such, early detection of developmental delays not associated with disability should also be investigated.”

Point 2: MATERIAL and METHODS. Line 98 and 99 state that only one assessor scored the Motor Optimality Score-Revised. Does this also apply to GMA or were several assessors involved? A more detailed description of how the MOS-R assessment and the GMA were done will be needed. Even if the reliability of MOS-R is described to be high only one assessor must be regarded as a weakness of this study.  

Response 2: We have added detail about the scoring of the GMA to page 4, line 106- : The GMA component of the MOS-R was scored by two advanced-certified raters and any disagreements settled through review between three raters. 

A limitation has been added to page 12, line 339-341: “Finally, due to resource constraints, only one assessor was available to score the MOS-R, however, previous studies have indicated high inter-rater reliability.[26]”

Point 3: Line 115: Did any of the infants have exaggerated fidgety movements, or is that regarded as the same as abnormal fidgety?

Response 3: Reply: Exaggerated fidgety movements are classified as abnormal fidgety movements. We have added clarification on the description of fidgety movements, including the sporadic and abnormal classifications on page 2, line 54-56: Fidgety movements are classified as normal, sporadic (observed only occasionally), absent (not seen at all) or abnormal (exaggerated with larger amplitude).

Point 4: RESULTS.  Line 196: What about the infants with sporadic fidgety movements?

Respnose 4: Reply: sporadic movements were absorbed into “absent fidgety” movement classification as there is no change in scoring between absent/sporadic movements. Changes made to page 4, line 120-121: GMA was classified as absent/abnormal (MOS-R, first subsection score 1 or 4, with sporadic movements classified as “absent”)